# Intercropping—A Low Input Agricultural Strategy for Food and Environmental Security

**Sagar Maitra [1]**, **Akbar Hossain [2],\***, **Marian Brestic [3,4],\***, **Milan Skalicky [4]**, **Peter Ondrisik [5]**, **Harun Gitari [6]**, **Koushik Brahmachari [7]**, **Tanmoy Shankar [1]**, **Preetha Bhadra [1]**, **Jnana Bharati Palai [1]**, **Jagadish Jena [8]**, **Urjashi Bhattacharya [7]**, **Sarath Kumar Duvvada [1]**, **Sagar Lalichetti [1]** and **Masina Sairam [1]**

1   Centurion University of Technology and Management, Odisha 761211, India; sagar.maitra@cutm.ac.in (S.M.); tanmoy@cutm.ac.in (T.S.); preetha.bhadra@cutm.ac.in (P.B.); jnana@cutm.ac.in (J.B.P.); sarathkumarduvvada010@gmail.com (S.K.D.); lalichetti.sagar@cutm.ac.in (S.L.); sairammasina52@gmail.com (M.S.)
2   Bangladesh Wheat and Maize Research Institute, Dinajpur 5200, Bangladesh
3   Department of Plant Physiology, Slovak University of Agriculture, Nitra, Tr. A. Hlinku 2, 949 01 Nitra, Slovakia
4   Department of Botany and Plant Physiology, Faculty of Agrobiology, Food and Natural Resources, Czech University of Life Sciences Prague, Kamycka 129, 165 00 Prague, Czech Republic; skalicky@af.czu.cz
5   Department of Environment and Zoology, Slovak University of Agriculture, Nitra, Tr. A. Hlinku 2, 949 01 Nitra, Slovakia; peter.ondrisik@uniag.sk
6   Department of Agricultural Science and Technology, School of Agriculture and Enterprise Development, Kenyatta University, P.O. Box 43844, 00100 Nairobi, Kenya; hgitari@gmail.com
7   Bidhan Chandra Krishi Viswavidyalaya, Mohanpur 741252, India; brahmacharis@gmail.com (K.B.); urjashibhattacharya@gmail.com (U.B.)
8   Indira Gandhi Krishi Viswavidyalaya, Raipur 492 012, India; jagadish759020@gmail.com
\*   Correspondence: akbarhossainwrc@gmail.com (A.H.); marian.brestic@uniag.sk (M.B.)

**Abstract:** Intensive agriculture is based on the use of high-energy inputs and quality planting materials with assured irrigation, but it has failed to assure agricultural sustainability because of creation of ecological imbalance and degradation of natural resources. On the other hand, intercropping systems, also known as mixed cropping or polyculture, a traditional farming practice with diversified crop cultivation, uses comparatively low inputs and improves the quality of the agro-ecosystem. Intensification of crops can be done spatially and temporally by the adoption of the intercropping system targeting future need. Intercropping ensures multiple benefits like enhancement of yield, environmental security, production sustainability and greater ecosystem services. In intercropping, two or more crop species are grown concurrently as they coexist for a significant part of the crop cycle and interact among themselves and agro-ecosystems. Legumes as component crops in the intercropping system play versatile roles like biological N fixation and soil quality improvement, additional yield output including protein yield, and creation of functional diversity. But growing two or more crops together requires additional care and management for the creation of less competition among the crop species and efficient utilization of natural resources. Research evidence showed beneficial impacts of a properly managed intercropping system in terms of resource utilization and combined yield of crops grown with low-input use. The review highlights the principles and management of an intercropping system and its benefits and usefulness as a low-input agriculture for food and environmental security.

**Keywords:** food; environment; intercropping; security; sustainability

## 1. Introduction

Agriculture plays a significant role in most of the developing countries. But due to the increased population and development of urban clusters along with industrial growth in the developing world, there is shrinkage in the availability of land for farming because of its

non-agricultural uses. Under these circumstances, the adoption of high-intensity cropping systems may be the viable option to increase agricultural productivity and production as a whole [1,2]. Agriculture is a tradition and heritage in most countries. Traditional farming practices are evidenced around the world with the growing of crop mixtures which is nothing but a form of mixed cropping or intercropping. The farming systems of ancient periods in different corners on the planet are documented as having grown crop mixtures which were nurtured by the people for a long time [3]. Early civilizations evidenced the use of intercropping that might be in a different form. South Asian subcontinent experienced growing of diversified crops with environmental diversity [4–6] since the Indus Civilization (c. 2600–1900 BC) in the form of mixed cropping or multi-cropping or intercropping [7,8]. Furthermore, the intercropping system was well known in Greece since about 300 B.C. which indicated growing of cereals and pulses [9] in which pulses were planted at different times during the growing season of cereals like wheat and barley [10]. Traditional mixed cropping has enough potential to contribute as much as 15–20% in the food basket of the world [11]. In Latin America, maize-based intercropping is very common [12]. In Africa, 89% of cowpeas and in Colombia 90% of beans are growing in mixed stands; however, in Malwai intercropping is very common and occupies 94% of cultivated land [13].

Various types of intercropping were known and presumably employed in ancient Greece about 300 B.C. Theophrastus, among the greatest early Greek philosophers and natural scientists, noted that wheat, barley, and certain pulses could be planted at various times during the growing season often integrated with vines and olives, indicating knowledge of the use of intercropping [9]. In tropical countries, intercropping is generally observed with food crop production, but much emphasis has been given in forage production in the temperate world for fulfilment of the high demand for animal feed [14].

Intensive mono-cropping is less complicated for big-scale farmers with the fullest utilization of machines, while smallholder farmers do not have steady access to markets and only produce enough food for their family members under subsistence farming. Intercropping ensures their livelihood. Therefore, intercropping is mainly practiced on small farms. Moreover, intercropping is known to produce stable yields from diversified crops with less use of inputs for nutrient supply and plant protection, focusing on sufficient food under healthy environmental conditions. In organic agriculture, intercropping is useful because less incidence of pest, disease and weed occurs and soil fertility is maintained [15]. During the recent period, the system approach in agriculture has drawn more attention from researchers. A system is comprised of different constituents that are high with interaction among them. The system approach targets greater use of available resources resulting in production sustainability and enhancement of intensity. The cropping systems with a more intense focus on raising of crops and varieties or hybrids having tolerance to biotic and abiotic stresses, capacity to replenish soil for sustainable production and express higher yield. Developing suitable cropping systems is an enormous job for achieving potential yield under any agro-climatic conditions [16]. Actually, some factors like resource availability and management are mainly considered to evolve a cropping system. The competence of a cropping system rests on the productivity of crops, time duration and requirement of land [17,18].

Modern agriculture based on supply with high energy and fossil-fuel-based inputs that are commonly known as Green Revolution Technologies has resulted in a remarkable enhancement in crop yields, but once this flourish was achieved agricultural sustainability disappeared [19]. The modern farming systems infer monoculture, replacing biodiversity with few crops and a limited number of cultivars in vast areas. On the other hand, on-farm biological diversity is maintained by the traditional farmers of developing nations and mixed cropping, intercropping and agroforestry patterns are prominently observed. These farming systems offer the ability to grow a variety of crops, assured production, efficient use of resources, less chance of crop damage by pests and diseases and proper use of the human workforce with a standard income [14]. Different crops grown in an intercropping system may or may not be seeded or harvested simultaneously, however,

the crops remain in the same field for a major part of the crop periods. There are various types of plant species that can be included in intercropping, namely, annual crops like cereals, legumes, oilseeds, fodder crops and so on. Low-input and energy-efficient crop production systems are no doubt attractive for sustaining agricultural productivity [11,20], but, modern agriculture imposed less diversified crop production system with the use of high energy inputs and chemicals. Agricultural sustainability can be achieved by creating on-farm diversity and proper use of natural resources with greater ecosystem services [21,22]. Biological diversity in a crop-growing environment can be improved by a recurrent succession of crops in sequential cropping or intercropping systems [11]. Actually, modern agriculture increased crop yields but developed issues related to agricultural sustainability [23]. An intercropping system has enough potential to enable sustainability in agriculture by crop diversification, efficient resource management and soil fertility restoration. The review focuses on essential aspects of intercropping systems as low-input management practices for food and environmental security with agricultural sustainability.

## 2. Intercropping as Low-Input Agriculture

An economically viable agriculture production system demands a supply of sufficient quantity of inputs. The success of green revolution (GR) in the second half of the previous century greatly focused on the supply of essential inputs and so green revolution technologies (GRTs) were more commonly known as supply-driven technologies. As per the concepts of GRTs, important inputs used in agriculture are high-yielding varieties (HYVs), fossil fuel-based high-energy chemical fertilizers, assured irrigation, use of sufficient plant protection chemicals and so on and interestingly all these inputs need high energy. On the other hand, intercropping systems need comparatively fewer energy inputs like fertilizer, plant protection chemical requirements are less, and diversification of crops is greater creating functional diversity resulting in less pest-disease incidence. Moreover, there is the creation of soothing microclimate with less evaporation [24,25]. Combination of deep-rooted and shallow-rooted crops create the options of bio-irrigation and after all, legume crops in association with non-legumes favour adjustments of nutrients by benefitting non-legumes in the form of nitrogen fixation. The legumes, furthermore, create a congenial environment for harbouring different beneficial microorganisms favouring higher ecosystem services. The cumulative impacts of all factors are reflected in the intercropping system and thus the intercropping system can be considered as a low input agriculture practice with higher output in terms of higher farm output and agricultural sustainability.

## 3. Concept and Goal of Intercropping

Cultivation of two or more crop species concurrently as they coexist for a significant part of the crop cycle is known as intercropping and it is also sometimes termed as polyculture or mixed cropping [14]. The component crops are neither seeded at the same time nor harvested, but they remain simultaneously in the field for a major portion of the growth periods of component crops. Intercropping is, in general, comprised of the main crop and one or more companion crops, where the production of the main crop is the prime goal. Intercropping is actually the value addition of the cropping system which can ensure higher productivity, efficient use of resources, and more income [26–28].

The history of the adoption of intercropping is not known, but ancient civilizations witnessed cultivation of crop mixture. Intercropping is still adopted in developing countries and it is also observed that intercropping began disappearing from many areas with the advent of high energy-based modern agriculture. The shift from polyculture to so-called 'modern monoculture' was driven primarily by commercialization and specialization of industrial agriculture along with the involvement of chemical-based inputs and assured irrigation. Increasing interests in sustainable crop production and ecological issues have distracted consideration back to polyculture as a path of efficient use of available resources with as much care as possible for ecology and leading towards agricultural sustainability. Since the historical period, intercropping activities were noted in different countries of

the world with various crop mixtures with cereal mixtures found commonly in temperate regions [12]. Nonetheless, intercropping gained importance and is widely practiced in tropical regions because of extensive genetic diversity in terms of crop choice as well as cropping systems [29–31]. Furthermore, the decline in temperature and rainfall is inversely proportional to adoption in a variety of intercropping systems [32]. With the adoption of industrialized farming, intercropping started to disappear from different parts of developed and industrialized countries as monoculture became popular there. This drift was motivated by the use of high energy inputs, improved farm machinery and specialization and these were considered as the prime strategy for enhancing crop yield. This industrialized agriculture was successful with a single crop or commodity, but the question of higher system products, as well as agricultural sustainability, remained unanswered over time [33].

In intercropping, basic ecological principles are observed in the form of above and belowground diversity, competition, and facilitation, for production of crops [34]. Generally, if the polyculture system of crops is chosen with proper prerequisites, the yield output appears higher than pure stands of individual crops. Moreover, in the intercropping system, different resources are better used by crops from a common pool compared to pure stands of the respective crops which result in greater productivity [34,35]. An intercropping system assures more coverage of the ground area by the canopy of crops, more transpiration takes place by the foliage which may create a cooler microclimate, and this facilitates the ability to minimize the soil temperature [36]. Under moisture stress conditions, in intercropping systems, crops use available water in the form of soil moisture and this microclimate provides a soothing effect at the canopy level of crops [37]. Generally, in intercropping, morphologically dissimilar crops are chosen with different growth habits, so available resources are efficiently utilized and the ultimate gain is the conversion into the crop dry-matter production or crop yield [10]. Different factors like choice of crops and cultivars, sown proportions and agronomic management including water and nutrients and the competitive ability of crops can affect the performance as well as the success of intercropping systems.

## 4. Types of Intercropping

Intercropping is the raising of two or more crops together as they coexist for some time on the same land. The spatial and temporal crop intensification is done in intercropping and it may be of different combinations of annual and perennial crops as per the choice of the farmers and suitability to the growing conditions [38]. Furthermore, in intercropping, competition is noted among the component species grown during the entire crop period or a part of growing duration for available resources. Different types of intercropping systems are adopted in various countries which can be grouped into the following [39].

### 4.1. Row Intercropping

The row intercropping is raising of one or more crops sown in regular rows, and growing intercrops in a row or without row at the same time. The row intercropping is a usual practice targeting maximum and judicious use of resources and optimization of productivity [40].

### 4.2. Mixed Intercropping

In mixed intercropping, two or more crops are grown together without any definite row proportion. Sometimes it is also referred to as mixed cropping [41]. In pasture-based cropping system, grass-legume intercropping is an ideal example of mixed intercropping [42]. The mixed intercropping is commonly observed to fulfil the requirement of food and forage where the land resource is a limiting factor [43]. Furthermore, a review work clearly described perennial polycultures as an agroecological strategy in cropping system with enough potential for the sustainable intensification of agricultural systems spatially and temporally [44].

### 4.3. Strip-Intercropping

The strip-intercropping is a type of intercropping where two or more crops are cultivated together in strips on sloppy lands. Strip intercropping is known to enhance greater radiation use efficiency in marginal and poor lands [45]. A combination of soil conserving and depleting crops are taken in alternate strips running perpendicular to the slope of the land or the direction of prevailing winds. An important objective of strip cropping is the reduction of soil erosion and harvesting of yield output from poor lands.

### 4.4. Relay Intercropping

Relay intercropping is raising two or more crops at a time during a portion of the growing period of each. In this system, the second crop is seeded when the first crop completes a major part of its life cycle and reaches reproductive stage or close to maturity but before harvest. The areas with limitation of time and soil moisture are more appropriate for relay cropping [46]. Before harvesting of the preceding crop, the next crop is sown and both the crops remain in the field for some period of their cycle. However, the succeeding crop yields less compared to normal sowing in sequential cropping and more seeds of the succeeding crop are required to obtain a good stand.

## 5. Crop Geometry in Intercropping

The proportional row arrangement of different crop components crops in an intercropping system ascertains advantage or disadvantage of intercropping compared to pure stands of the respective crops [45]. Based on the arrangement of rows and the proportion of crops sown, the intercropping system is grouped into the following two categories.

### 5.1. Additive Series

In the additive series, intercrops are added in 100% population of the base crop. The crop sown with 100 per cent density as seeded in the pure stand is called 'base crop' and another crop is termed as 'intercrop'. In this system, 'intercrops' are sown within the row space of the 'base crop' and sometimes planting geometry of the base crop is modified to create space for intercrops (for example, paired row planting). The proportion of 'intercrop' (not base crop) mostly remains less than its optimal population in sole cropping. As the addition of 'intercrop' has been considered in 100% population of the 'base crop' in additive series, the land equivalent ratio (LER) is mostly observed as more than unity indicating yield advantage. Yield advantage gives extra monetary return and so it is considered as an efficient intercropping system and commonly preferred by small landholders in developing countries.

### 5.2. Replacement Series

In the replacement series of intercropping, the crops grown together are known as component crops or intercrops. Here, one component crop is introduced by the replacement of the other crop. In the replacement series of the intercropping system, no crop is sown with its fullest population as seeded in respective sole cropping. In this system, a definite proportion of a crop is sacrificed and the component crop is introduced in that place. Sometimes to obtain yield advantage from replacement series of intercropping, plant population is increased compared to their density adopted in the pure stands [47]. In such an intercropping system, competition among species is relatively less than additive series.

## 6. Consideration for Choosing the Intercropping System
### 6.1. Crop Choice

The crop choice is an important consideration concerning the growing situation, crop environment of a locality, suitability of the crop as well as demand and availability of a particular variety [41]. The appropriate crop mixtures show complementarity among the species cultivated and yield advantage is observed. Furthermore, in a study in West Bengal India, by Maitra et al. [48] it was noted that finger millet, when intercropped with

red gram or groundnut (4:1 ratio), expressed benefits in terms of more net return and benefit: cost ratio than other combinations considered like finger millet + green gram and finger millet + soybean in rainfed conditions. Similarly, Fan et al. [49], recorded more grain output of faba bean + maize; but the yield of fava bean was less in a faba bean + wheat intercrop combination. These are the examples of the importance of crop choice in yield enhancement while selecting the crop in intercropping, generally, crop morphological and physiological characters are considered. For example, combinations of deep and shallow-rooted crops (like finger millet and green gram) or crops with tall and dwarf canopy (like maize + groundnut) are preferred for better utilization of the available resource. Intercropping in maize is very common and legumes are preferably chosen in maize-based intercropping system. In different intercropping studies, it was noted that maize + legume combination registered more yield with greater use of resources [28,48] which are the primary goals of the intercropping system. There are several crop species which may be considered in intercropping, like annuals, perennials and mixture of the both. In alley cropping, a type of agroforestry, perennials are chosen in hedgerows and annual crops are cultivated in alleys. Moreover, the growing period and time of peak demand for resources of different crops selected in the intercropping system is also important to get maximum benefit.

*6.2. Crop Maturity*

Crop maturity is an important factor for the choice of crops in an intercropping system. The crops preferred in intercropping combination should be of a different kind in terms of their grand growth period, otherwise, there may be a chance of inter-species competition for required resources if it coincides. The complementarity among the species is desirable to obtain the benefits of an intercropping system which are reflected as system productivity. Hence, the crops chosen should be of different duration with dissimilarity in the form of growth and morphology as they can exhibit complementarity among themselves. As an example, it may be stated that maize has been considered a suitable cereal species and also treated as a base crop in the intercropping system in association with preferably dissimilar legumes of shorter lifecycle [48]. Green gram or black gram is of short duration pulse crops when grown as intercrop in association with the base crop of maize, pulses enter into the reproductive stage before maize reaches to the knee-height stage (approximately 6–7 weeks after planting) and thus least competition is observed among the crops. The result of such combination expresses a higher level of mutual benefits in the expression of crop yields of individual species.

*6.3. Planting Density*

To obtain optimal yield output it is necessary to maintain proper plant stand. But in replacement series, there will be the reduction of plant population of crop species in comparison to sole crops, whereas in additive series, the base crop gets a similar plant stand and other crops that are accommodated may or may not occupy areas like sole cropping. Furthermore, paired-row geometry of planting of the base crop is beneficial because more space for intercrops is created. Sometimes in replacement series of intercropping system, population density is enhanced compared to the pure stand of individual crops to achieve higher system productivity with greater leaf area index (LAI) [50]. In an intercropping system with base crops like maize, cotton, sugarcane and so on, paired row planting in intercropping is commonly practiced [28,51]. The crops with various durations and different growth habit with peak demand for nutrients are chosen to minimize competition among the species.

*6.4. Planting Time*

In intercropping systems, sowing/planting time of component crops may or may not vary as in relay intercropping system. Intercrops are introduced when the base crop reaches close to its maturity or complete a major period of its growth. The competition among

the species is much less in relay intercropping. In south Asian countries, relay cropping of pulses and oilseeds is very common in rainy season rice and by utilizing residual soil moisture and nutrients relay crops yield satisfactorily. When the crops are sown together in intercropping, preferably crops with a different type of growth habit are chosen. For example, in maize-based cropping systems short duration green gram or black gram if sown completely the major part of their growth before maize reaches its peak demand stage. As maize is used as fodder also and maize-legume fodder mixed cropping system is common in different countries. Under this situation, dry matter or biomass production is the ultimate target and competition among crop species does not influence the forage yield. With grain crop maize, legumes generally yield quite reasonably because of wider spacing adopted in maize sowing.

## 7. Management of Intercropping

Under diverse conditions, farmers adopt different intercropping practices and thus the intercropping system itself becomes complex [52]. There is no doubt that the management of the intercropping system is difficult in terms of requirements of more human labour. But considering the multiple benefits as well as agricultural sustainability intercropping system may be considered as one of the suitable options for food and livelihood security of small farmers.

### 7.1. Seed-Bed Preparation

A seed-bed is prepared before sowing by physical manipulation of soil and suitable tillage is required for different crops [53]. In intercropping when two or more crops with dissimilar morphological characters are sown together, uniform bed preparation may not be ideal for different crops and it greatly depends on crops [54]. For example, deep-rooted crops need deep tillage, while cereals require shallow tillage. The crops with small seeds (like mustard, sesame and jute) require pulverized soil and fine beds. Some crops are sown on ridges (cotton, maize) whereas others such as green gram, black gram and mustard prefer flat-beds. In additive series, the bed is prepared as per the requirement of the base crop. In maize + green gram/black gram intercropping, generally, crops are sown on a flat-bed [28]. In a sugarcane-based intercropping system, sets are planted in furrows, but intercrops are sown on ridges [50,55].

### 7.2. Varieties

The varieties of crops chosen in intercropping should have some desired characteristics as the highest level of complementarity and least competition occur [54,56]. The crop varieties are required to be photoperiod insensitive as these can be cultivated at any time of the year [57,58]. The short duration sorghum hybrids like CSH 17 (103 days) and CSH 23 (105 days) are suitable for intercropping long-duration pigeon pea varieties like GAN 1 and WRP 1 (both are of 160–165 days). The varieties selected for intercropping should have some morphological and physiological characteristics like thin leaves, less branching and be tolerant to shading.

### 7.3. Sowing and Plant Stand

In the intercropping system, modification or alteration is done in planting geometry, spacing and thus plant stand [59]. Paired row planting is one of the modifications, where two rows of the base crop are sown in close spacing followed by a wide gap between two pairs and spaces between two pairs are used to accommodate intercrops. In additive series of an intercropping system, for widely spaced crops like cotton, maize and red gram, paired row planting is beneficial and higher total yields are obtained from the crops grown in the mixture. Furthermore, the base crop population is maintained equal to the plant population of sole cropping. Paired row cotton yielded more than uniform row cotton in entisols of Sundarbans when intercropped with a single row of green gram and groundnut and paired row planting registered higher monetary advantage than uniform

row planting of cotton with the same combination of legumes [51]. In a pearl millet + green gram intercropping system, paired row planting resulted in more yield than uniform row planting, and intercropping of paired row maize + pigeon pea performed well in the southern dry zone of Karnataka [60]. In replacement series after sowing of a crop with some uniform rows, replacement is done by another crop of some rows and row proportion is determined mainly by the farmers or as per the recommendations. Sometimes closer spacing within rows is also followed to accommodate more number of plants and generally a greater number of plants is accommodated in intercropping. Other factors related to optimal and uniform plant stands include seed treatment, bed preparation, sowing at the proper depth and so on, are maintained as per the standard procedure. Further, gap filling is an important operation in drylands, which can also be taken care of to obtain the desired plant population.

### 7.4. Fertilizer Application

The nutrient removal by the crop is greater because of more dry matter production or biological yield in intercropping. In a cereal–legume combination of intercropping, legumes use less N from the soil and it may be either from inherent soil fertility or in the form of applied fertilizer. On the other hand, cereals are more N demanding and use a major portion of applied N. Nevertheless, legumes initially use P for better nodulation, but after nodulation, the root exudate of legumes and other rhizospheric micro-organisms make P available to both legumes and the companion cereals. In N-deficient soils, legume fixes a considerable quantity of N, but when sufficient N is supplied as fertilizer, biological N fixation is reduced. Moreover, sufficient supply of N fertilizer promotes the growth of cereals because cereals are more aggressive in nature and the growth of legumes is suppressed. Considering the above, it is advisable to apply N as basal and topdressing to cereal rows and P and K to the whole plot. In the molybdenum-deficient soils, the micronutrient application should be done as basal or by foliar spray, because sufficient molybdenum enhances nodulation as well as biological N-fixation by legumes [61]. The mutual benefit or complementarity observed in the cereal–legume combination is the result of below-ground chemical and biological processes which can assure the availability of some micro-nutrients like iron and zinc [62].

### 7.5. Water Management

Water is the most valuable resource and it has a great impact on national development which needs special attention for enhancing higher output from land, better efficiency, increased earnings, and maintaining the ecological balance. Sustainable agricultural production can be achieved if different natural resources, more especially water resource are utilized efficiently [63]. Management of water resource in an integrated manner is a concept where water is used judiciously and in agriculture the optimal use of irrigation water for enhancing water productivity per unit area. There is no basic difference in the management of water in multiple cropping, sequential cropping and intercropping so as to provide irrigation for crop needs. However, when two dissimilar types of crop are taken into consideration in intercropping, special management techniques are to be followed in intercropping. Moreover, in additive series of intercropping, the combined plant population appears as more than 100%, which means soil moisture use in the form of evapotranspiration increases. Under rainfed, moisture-stress conditions and dryland situations, the extra need for water for crop mixture in intercropping will be an additional burden to resource-poor farmers. On the other side, more coverage of ground area restricts evaporation loss and, generally, it is observed that the water needs of intercropping do not exceed that of pure stands. Nyawade et al. [25] in upper midland of Kenya observed that in rainfed potato + legume intercropping system more LAI was noted which was indicative of more coverage of the ground area that lowered soil temperature by 7.3 °C at 0–30 cm depth and ultimately increased soil water content and crop water productivity than sole cropping of potato. In an oasis of arid north-west China, in wheat + maize intercropping

system, alternate irrigation exhibited higher water use efficiency (WUE) than conventional irrigation of either of the crops [64]. In China, in a maize + soybean intercropping system maximum water use efficiency and water equivalent ratio were noted with 40:160 cm planting geometry using 200 cm bandwidth [65]. Similarly, in another experiment, [66] noted that strip-intercropping with pea and maize in China showed complementarity in sharing water and both the crops gave more grain yield by 25% and enhanced water use efficiency by 24% than solely maize. However, in irrigated conditions when sufficient water is available, the crop combinations may suffer due to the difference in water requirements. For example, in cotton + green gram / black gram intercropping, irrigation should be given to cotton at an interval of two to three weeks, but these legumes do not require the same frequency and well as quantity and legumes may suffer due to excess water. In most of the intercropping systems, it is better to schedule irrigation by following IW/CPE ratio. Bio-irrigation is another beneficial phenomenon by which shallow-rooted crops obtain support from deep-rooted ones in intercropping under limited moisture conditions. The experimental results clearly indicated that deep-rooted pigeon pea played the role of bio-irrigators and shared moisture for shallow-rooted finger millet [67,68].

### 7.6. Weed Management

In intercropping systems, chemical herbicide application is difficult once crops have emerged particularly when a combination of dicotyledonous and monocotyledonous plants are chosen in combination [69]. However, as the greater portion of the land area is covered by the crops in intercropping, there will be fewer weeds. Fast-growing crops like mung and black gram under intercropping cover maximum land area and suppress weed growth. The weed suppression ability in the intercropping system depends on some factors like selection of crop, the genotypes used, plant population, the ratio of crops considered in the intercropping and spatial arrangement, fertility and soil moisture. Mostly hand weeding is practiced in intercropping. Weed control by the application of chemical herbicides is difficult as most of the herbicides are crop-specific. The more complex the intercropping system, the less likelihood of a finding of herbicides. Earlier Reddy [70] mentioned that Isoproturon (1.0 kg ai ha$^{-1}$) was effective in intercropping wheat + chickpea and wheat + mustard. Likewise, Alachlor (1.5 kg ai ha$^{-1}$) was beneficial in maize + cowpea and sorghum + black gram intercropping systems. Further, Butachlor (1.25) resulted in successful control of weeds in maize + mung intercropping system [70].

Furthermore, allelopathy may reduce weed population. Researchers noted less weed growth in different intercropping combinations like wheat and chickpea [71], maize + legume [72] and so on. Paired row planted maize + 2 rows of soybean/2 rows of sesame reduced weed growth in Nagaland [73]. Experimental results showed that intercropping of maize + soybean and maize + cowpea significantly reduced weed growth than sole cropping and the pre-emergence application of alachlor 2 kg ha$^{-1}$ or metolachlor 1.0 kg ha$^{-1}$ controlled weeds successfully [74]. Similarly, maize + cowpea and maize + black gram suppressed weeds more than pure stands of maize, but maximum weed control efficiency and yield of maize were noted by the herbicide Pendimethalin at 0.75 kg ha$^{-1}$ and mechanical hoeing at five weeks after sowing [60,75]. Moreover, in intercropping systems available resources are efficiently utilized by two or more crops in combination and thus weeds do not get exploitable resources and growth of weeds is suppressed [76]. In a study, Divya et al. [77] noted that less density of grassy weeds and sedges were observed when runner bean (*Phaseolus coccineus* L.) was intercropped in maize than pure stands of maize.

### 7.7. Pest and Disease Management

The insect-pest population is regulated by the intercropping system itself. In marginal farming mixed cropping is chosen because of the low incidence of insect pests [78]. The crop mixture attracts beneficial insects which have the potential to maintain the harmful pest population below the threshold level [22]. Researchers noted less incidence of insect pests in intercropping and the presence of natural enemies as observed in maize-based

intercropping with beans and cowpea [79]. In Nigeria, the weboorm (*Antigostra* sp.) caused less damage to sesamum when intercropped with sorghum [80]. Polyculture reduced the population of *Empoasca krameri* by 26% in beans and *Spodoptera* by 14% in maize in intercropping compared with their pure stands [81]. When cowpea was intercropped with cotton, sucking pest population was reduced [82] and stem borer (*Chilo zacconius*) and green stink bug (*Nezara viridula*) were checked when upland rice + groundnut was intercropped [83]. Adoption of integrated pest management is advisable to keep the pest population below the threshold level. In general, adoption of cultural measures reduces the chance of pest attack and when the attack is noticed mechanical, biological and chemical methods should be applied for the protection of crops. Intercropping also checks plant diseases by creating a functional diversity which limits the population of harmful micro-organisms [84]. For the management of diseases, also the application of the integrated approach is the best. Management of seed-borne pathogens can be done by treating seeds with chemicals or bio-fungicides. Removal of diseased plants will reduce the inoculum source. Planting of disease-resistant cultivars is also a suitable measure to reduce disease incidence.

## 8. Indices for Measuring the Efficiency of Intercropping

In intercropping systems, most of the competition studies have examined growing two crop species and also in a 'replacement series'. The component crops involved in the system may be related to each other in the manners mentioned below:

(i) Competitive: In this relationship, the output of one crop would be increased through the decline in the production of the other. This is also known as 'compensation'. Willey [85] referred to the two species as 'dominant' and 'dominated' species.

(ii) Complementary: This is another type of relationship in which an increase in output of one crop helps to bring about an increase in output of the other species. This is termed as 'mutual cooperation' [85] and is not very common.

(iii) Supplementary: In this case, the output of one crop may be increased without having any influence on the output of the other. This situation commonly occurs when the maturity of two crop species differ widely.

(iv) Mutual Inhibition: Mutual inhibition happens when the actual productivity of each component of crops harvested is less than the expected yield. The competitive and supplementary relationship is very common in different intercropping systems. The majority of research works carried out that for value assessment of variation among pure stand and the intercropping system was developed during the period from 1970 to 1980. Most remarkable was the proposal of the land equivalent ratio (LER) and afterwards, widespread application of the LER was noted to evaluate the performance of an intercropping system) [85–87]. Later various researchers reviewed these studies and validated the concept of LER [12,39]. The focus of these studies was mostly on the use of replacement series of intercropping (mainly with two crops) and productivity of intercropping is compared with pure stands of each crop species. A major problem is that additive series of intercropping the LER exhibits the combined value of base crop with 100% plant density and the additional value of intercrops which ultimately results in the combined LER value with more than unity [86,88,89]. However, researchers concluded that the derivation of LER values is the concerned researchers' concern in estimating the efficiency of an intercropping system over pure stand [33].

There are also other concepts developed over time by different researchers to describe the competitive relationship and of them, some are also described below:

### 8.1. Land Equivalent Ratio (LER)

Willey and Osiru [86] gave the idea of the LER and it is described as the proportionate land area required under a pure stand of crop species to yield the same product as obtained under an intercropping at the same management level [48]. The LER of intercropped



plots are estimated for each component crops separately by adding the estimated total of two varieties; the LER of the sole crop is taken as unity (1). In a replacement series of intercropping with a combination of two crops at the ratio of 50:50, the LER can be calculated by the following expression.

$$\text{LER} = \frac{Yab}{Yaa} + \frac{Yba}{Ybb} = La + Lb \tag{1}$$

where, $Yab$ is the yield of "$a$" crop grown in association with "$b$" crop and $Yba$ is the yield of "$b$" crop grown in association with "$a$" crop. $Yaa$ and $Ybb$ represent the yields of "$a$" and "$b$" crops grown in a pure stand, respectively.

The modified formula for any other situation is:

$$\text{LER} = \frac{Yab}{Yaa \times Zab} + \frac{Yba}{Ybb \times Zba} = La + Lb \tag{2}$$

The LER denotes the benefits of an intercropping system to utilize the resources as against their pure stands [88]. The LER value greater than unity (1.0) indicates the advantages of the intercropping system [39] and less than one (1.0) is considered as a poor performance of the intercrops [90]. The LER value of some intercropping system with major crops is presented below (Table 1).

**Table 1.** Land equivalent ratio (LER) in intercropping systems.

| Intercropping System | Ratio | LER | Country | References |
|---|---|---|---|---|
| Sorghum + Sesbania | 2:1 | 1.06 | Syria | [91] |
| Wheat + Faba bean | 1:1 | 5.24 | UK | [92] |
| Sorghum + Cowpea | 2:1 | 1.08 | Nigeria | [93] |
| Wheat + Mustard | 1:1 | 1.46 | Bangladesh | [94] |
| Wheat + Fenugreek | 1:3 | 1.4 | Pakistan | [95] |
| Wheat + Maize | 1:1 | 1.19 | China | [96] |
| Sorghum + Soyabean | 1:1 | 1.40 | Nigeria | [97] |
| Pearlmillet + Soybean | - | 2.77 | Nigeria | [98] |
| Sorghum + Ground nut | 1:1 | 2.10 | Ethiopia | [99] |
| Maize + Soybean | 1:1 | 1.54 | Nigeria | [100] |
| Maize + Groundnut | 2:2 | 1.42 | Ghana | [101] |
| Maize + Potato | 1:2 | 1.58 | Ethiopia | [102] |
| Maize + Garden pea | 1:2 | 1.56 | Bangladesh | [103] |
| Maize + Groundnut | 2:2 | 1.82 | India | [28] |
| Maize + Soybean | 2:2 | 1.90 | China | [104] |
| Wheat + Lentil | 2:2 | 1.34 | India | [105] |
| Potato + Dolichos | 1:2.4 | 1.24 | Kenya | [106] |
| Potato + Vetch | 1:2 | 1.75 | Kenya | [16] |
| Pearlmillet + Green gram | 1:1 | 2.03 | India | [107] |

### 8.2. Area Time Equivalent Ratio (ATER)

The LER emphasizes on the only land area without considering the time factor for which the crop occupies the field. As time factor is not a part in the LER, researchers needed another expression considering the field occupancy by the crops in an intercropping to correct this constraint of the LER. Hiebsch [108] developed the concept of Area Time Equivalent Ratio (ATER) in which the duration of crops (starting from seeding to harvest) was considered. The ATER is calculated by the following formula.

$$\text{ATER} = \frac{(RYc \times tc) + (RYp \times tp)}{T} \tag{3}$$

where, RY = Relative yields of crop species "c" and "p" = Yield of intercrop ha$^{-1}$/ Yield of sole crop ha$^{-1}$, t = duration (in days) for species "c" and "p" and T = duration (in days) for the intercropping system. However, the LER generally overemphasizes and the ATER

undervalues the land-use efficiency [27]. Researchers revealed the advantageous ATER values in different intercropping systems (Table 2).

**Table 2.** Area time equivalent ratio (ATER) in maize-legume intercropping systems.

| Intercropping System | Proportion | ATER | Country | References |
|---|---|---|---|---|
| Cotton + Cowpea | - | 1.13 | Pakistan | [109] |
| Lupine + Wheat | 75% + 100% | 1.31 | Ethiopia | [110] |
| Maize + Soybean | 2:6 | 1.32 | India | [111] |
| Maize + Black cowpea | 2:2 | 1.51 | India | [112] |
| Pearlmillet + Green gram | 2:1 | 1.25 | India | [113] |
| Wheat + Faba bean | - | 1.28 | Pakistan | [114] |
| Potato + Dolichos | 1:2.4 | 1.13 | Kenya | [106] |

### 8.3. Aggressivity

This was proposed by [115]. Aggressivity denotes a simple measurement of the quantity of the relative yield increases in crop species "a" than crop species "b". Aggressivity is expressed as "A". For a replacement series of treatment, aggressivity is measured by the following formula.

$$\text{A}ab = \frac{\text{Mixture yield of } a}{\text{Expected yield of } a} - \frac{\text{Mixture yield of } b}{\text{Expected yield of } b} \tag{4}$$

For other than replacement series of intercropping, the aggressivity is calculated by the following expression.

$$\text{A}ab = \frac{\text{Y}ab}{\text{Y}aa \times \text{Z}ab} - \frac{\text{Y}bb}{\text{Y}aa \times \text{Z}ab} \tag{5}$$

where, $\text{Y}ab$ = yield of crop "*a*" in intercropping system; $\text{Y}aa$ = yield of crop "*a*" in pure stand (sole cropping); $\text{Z}ab$ = sown proportion of crop "*a*" in intercropping; $\text{Y}ba$ = yield of crop "*b*" in intercropping system; $\text{Y}bb$ = yield of crop "*b*" in pure stand (sole cropping); $\text{Z}ba$ = sown proportion of crop "*b*" in intercropping.

The value of aggressivity (A) zero means none of the crops are considered as aggressive or both crops have equal competitive ability. But, when the aggressivity value becomes positive, then "a" crop is considered as aggressive or dominant over intercropped "b" crop. If 'A' value becomes negative, then intercropped "b" is considered as aggressive or dominant over "a" crop.

### 8.4. Competitive Ratio (CR)

In an intercropping system, competitive ratio (CR) denotes the competitive ability of the component species [116]. The CR expresses the number of times by which one component crop is more competitive than other [116] and CR actually represents the proportion of individual LERs of the crops considered in intercropping and also takes into account the ratio of the crops sown in a mixed stand. The CR can be calculated by the following formulae.

$$\text{CR}a = (\text{LER}a/\text{LER}b) \times \text{Z}ba/\text{Z}ab) \tag{6}$$

$$\text{CR}b = (\text{LER}b/\text{LER}a) \times (\text{Z}ab/\text{Z}ba) \tag{7}$$

where, CR*a* and CR*b* are indicative of the competitive ratios of the crop species "*a*" and "*b*" and LER*a* and LER*b* are the LERs of the crop species "*a*" and "*b*" respectively. Z*ab* is the sown ratio of species "*a*" in mixture with "*b*" and Z*ba* is the sown proportion of the species "*b*" in mixture with "*a*". If the value of CR is <1, there is a positive benefit and it means there is limited competition between component crops and therefore they can be grown as intercrops. If the CR value is more than one (CR > 1), there is a negative impact.

In this condition, the competition between intercrops in mixture is too high, and they are not recommended to grow as intercrops.

In Table 3, the values of CR of legumes appeared as >1, representing that legumes were more competitive than finger millet. Among the legumes, green-gram was found to be the least aggressive on affecting the growth of finger millet and thus it provided a balanced competition with finger millet.

**Table 3.** Competitive ratio (CR) of finger millet + legume intercropping (4:1) systems [27].

| Intercropping Systems | Competitive Ratio (CR) | |
|---|---|---|
| | **Finger Millet** | **Legumes** |
| Finger millet + Red gram (4:1) | 0.28 | 3.59 |
| Finger millet + Green gram (4:1) | 0.71 | 1.41 |
| Finger millet + Groundnut (4:1) | 0.58 | 1.73 |
| Finger millet + Soybean (4:1) | 0.68 | 1.48 |

*8.5. Relative Crowding Coefficient (RCC)*

In intercropping system, the relative crowding coefficient (RCC) indicates relative dominance of one component crop over another. The concept of RCC has been proposed by De Wit [117] and examined in detail by Hall [118,119]. It assumes mixture treatments from a replacement series. Each species in an intercropping system has its own co-efficient (K) which gives the measure of whether that species has produced more or less than expected yield. The RCC is calculated by the following formulae. For species "*a*" in a 50:50 mixture with species "*b*", RCC is measured as:

$$\text{Product of RCC (K)} = K_{ab} \times K_{ba} \tag{8}$$

where, $K_{ab} = (Y_{ab})/(Y_{aa} - Y_{ab}) = $ (Mixture yield of *a*)/(Pure yield of *a*—Mixture yield of *a*); $K_{ba} = (Y_{ba})/(Y_{bb} - Y_{ba}) = $ (Mixture yield of *b*)/(Pure yield of *b*—Mixture yield of *b*).

Furthermore, for a mixture differing from 50:50 proportion, RCC can be generalized as:

$$K_{ab} = (Y_{ab} \times Z_{ba})/(Y_{aa} - Y_{ab}) \times Z_{ab} \tag{9}$$

$$K_{ba} = (Y_{ba} \times Z_{ab})/(Y_{bb} - Y_{ba}) \times Z_{ba} \tag{10}$$

where K is the product of RCC; $K_{ab}$ and $K_{ba}$ are RCC for the crop species "*a*" and "*b*" respectively; $Y_{ab}$ = yield of crop "*a*" in intercropping, $Z_{ba}$ = sown proportion of crop "*b*" in intercropping; $Y_{aa}$ = yield of crop "*a*" in sole cropping, $Z_{ab}$ = sown proportion of "*a*" in intercropping and $Y_{ba}$ = yield of crop "*b*" in intercropping, $Y_{bb}$ = yield of legume "*b*" in the sole cropping.

When the RCC (K) value, that is, the product of two coefficients ($K_{ab} \times K_{ba}$) is more than one (>1), there will be yield advantage in the intercropping. When K appears as one (1), there is no yield advantage/disadvantage. However, when the value of K appears as less than one (<1), there is a competition between intercrops indicating disadvantage in intercropping.

## 9. Benefits of Intercropping

Agricultural sustainability is a countless task for all developing countries and more precisely for a populous nation like India to produce bumper for continuously increasing needs. The accessibility of farm-land is contracting each day because of other uses. In this context, one of the significant approaches to enhance farm output is the system approach which is considered a holistic approach too. A system is comprised of some inter-related and interacting components and system approach enhances the efficiency of use of available resources. Developing appropriate cropping systems based on agro-climatic conditions and resources is a huge task for realizing potential output. The consequence of a cropping system is estimated by the productivity of the crops grown by using the resources efficiently.

However, the latest concepts of agronomy suggest not only measuring the productivity of an individual and/or component crops of the system but another two dimensions, namely, time and space.

The cropping system consists of sequential cropping and intercropping. The potential of sequential cropping is already exploited, but the scope for achieving benefits from intercropping is still untapped. There are numerous reports concerning the beneficial outcomes and predominance of intercropping over the pure stand. A few specialists express their idea that intercropping is reasonable just for the smallholders who are engaged with subsistence cultivation. However, it has been observed in various parts of the world that polycultures in farming have enough potential to achieve agricultural sustainability because of diversification due to crop mixture [41,120,121]. Intercropping is beneficial in many ways as it assures greater resource use, reduction of the population of harmful biotic agents, higher resource conservation and soil health and more production and sustainable output of the system [2,48]. In an intercropping system, more than one crop is grown together on the same land and utilizes the soil nutrients, soil moisture, atmospheric $CO_2$ and sunlight. The resource conservation and soil health aspects are also positive effects of an intercropping system as it checks run-off, soil erosion and less nutrient loss from the soil [122,123]. Moreover, it facilitates soil fertility enhancement when small millets are intercropped with legumes and enables the diversity of beneficial soil microorganisms. In an intercropping system, complementarity among the species cultivated is very important for increasing crop yields.

In drylands, the intercropping system offers natural insurance against the failure of a crop. Different crops grown in an intercropping system require dissimilar agronomic management including post-harvest care. Mechanization is difficult in this situation and so more employment generation is created. In dryland regions, farming is not practiced throughout the year and unemployment of the farm workforce is an issue which can be minimized to some extent as intercropping is a labour-intensive practice. Cereal-legume combination provides food and nutritional security to smallholders of drylands. The population dynamics of different biotic agents namely, weeds, insect-pests and pathogen are changed. In small millet + legume intercropping system, cereal component gets benefit due to legume effect. Ultimately, less (chemical) inputs are involved in agriculture based on intercropping. Growing of two or more crops not only creates crop diversity but also makes the ecology favourable for predators. In other words, it may be stated that a better ecosystem service is achieved by the intercropping system which leads agriculture towards sustainability (Figure 1).

*9.1. Yield Advantage*

In the intercropping system, the same land area or unit area is provided to two or more crops in association and preferably more total yield is obtained from crops. The crop species grown in the mixture may show complementarity and less competition among crops result in certain yield advantage [85]. In additive series of intercropping, when additional crop or crops include the normal population of the base crop, there will be assured yield advantage [28]. In replacement series, yield advantage is also obtained and complementarity among the crop species matters [27,51,124]. The yield advantage in intercropping is measured by using some competition functions like relative yield total (RYT), relative value total (RVT), and monetary advantage and base crop equivalent yield may be considered. In an experiment, Mandal et al. [125] noted 5.48 t ha$^{-1}$ of maize equivalent yield (MEY) in maize and soybean (1:2) intercropping against 2.48 t ha$^{-1}$ in sole maize. Relative yield total (RYT) values of intercropping were higher than unity in different experiments indicating yield advantage [126]. Moreover, Manasa et al. [127] mentioned that MEY was 7.6 t ha$^{-1}$ when paired row maize was intercropped with groundnut (2:2) as against sole maize yield of 5.7 t ha$^{-1}$ and RYT was 1.47.

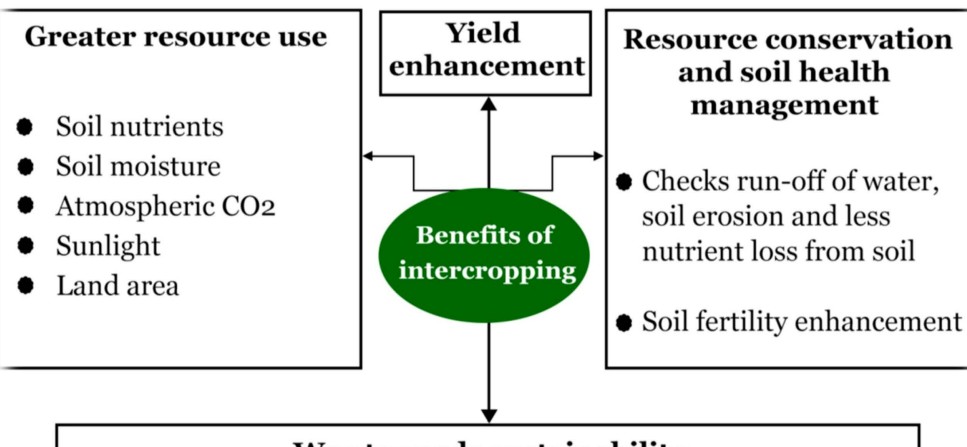

**Figure 1.** Benefits of intercropping system.

*9.2. Greater Use of Resources*

The intercropping system ensures a greater use of available resources [2], namely, land area, soil moisture, soil nutrients, sunlight and carbon dioxide used by a greater number of plants or crop species. Efficient utilization of these resources is reflected in higher biomass production and yield of crops. Higher yield is generally observed when the crops cultivated in association do not compete among themselves for the same resources. In other words, it may be stated that weaker intraspecific competition and greater complementarity among the crop species chosen are important for greater use of resources. Therefore, the combination of short and long duration crops or shallow and deep-rooted crops are preferred. The crops should be selected based on their resource use capability and competitive ability in time or space. Moreover, plant stand and planting geometry also influence resource use efficiency by crops in the mixture [48].

9.2.1. Soil Nutrients

The combination of cereal and legume in intercropping system triggers the soil fertility as legumes biologically fix N of about 80 to 350 kg ha$^{-1}$ [128]. Intercropping legumes changes the micro-organism colony dynamics in rhizosphere which facilitates increased mineralization of nutrients [59]. Different studies indicated that there were variations in physicochemical properties in the soil of rhizosphere by the adoption of the legume-based intercropping system [129] due to the addition of fresh organic matter and changed microbial population which increased the availability of organic carbon, nitrogen and phosphorus in the soil [130]. In the combination of cereal-legume mixed stands, root exudates alter the availability of nutrients [131]. Furthermore, the root exudates might be comprised of organic acids and enzymes. In a pea + barley intercropping system P, K and S accumulation were positively influenced which ultimately was reflected in the productivity of crops [132]. The intercropping system assures more utilization and removal of available soil nutrients. In the legume-based intercropping system, atmospheric N is fixed biologically and used by legumes as well as associated non-legumes as legumes share up to 15%

of N to cereals [133]. Generally, the combined biomass yield is more in intercropping and thus uptake by the crops is more. Not only N, but P and K uptake in intercrops were also more by 43% and 35% compared to pure stands [134] and such enhancement was due to more dry matter production. Enhanced P uptake was reported by different researchers under a varied intercropping system where legume was considered as a component, for example, pigeon pea + sorghum intercropping [135] and lupin + wheat intercropping [136]. Mobasser et al. [59] also suggested more P uptake in intercropping. In acid soils, P is a limiting factor and root of some crops like groundnut, cowpea, maize and beans secrete organic acids and phosphatases into the rhizosphere and enhance P availability [137]; hence intercrops grown in combination with these crops get more P. Furthermore, in acid soils, Al toxicity may be harmful to crops and the release of organic acids protects the roots from Al toxicity [138]. The soil microbiome plays a vital role in making soil nutrients available to crops and it happens more prominently in a cereal–legume intercropping system. Dai et al. [139] mentioned greater Fe foraging in maize + groundnut intercropping system in calcareous soil. In an intercropping system, the competition for nutrients among the crop species can be reduced by selecting appropriate crops with dissimilarity in nutrient needs, root morphology and time of peak requirements. Furthermore, use rhizobia inoculants in addition to the recommended fertilizer inputs applied in intercropping systems may boost the availability of nutrients to crops and thus enhance the productivity of the intercropping system [140]. In maize + pea intercropping system when crops were fertilized with low N with a higher density of maize, there was more corn yield in the oasis region of northwest China [141]. Actually, the mutual benefit or complementarity observed in mixed stands of cereal and legume in the intercropping system is the result of below-ground chemical and biological processes which can ensure the availability of some micro-nutrients like iron and zinc [62].

### 9.2.2. Available Soil Moisture

In intercropping systems, plants grown simultaneously use the common pool of soil moisture. The combination of shallow and deep-rooted crops is beneficial for efficient utilization of available soil moisture [24]. Chen et al. [66] observed that strip intercropping with shallow-rooted pea and maize showed complementarity in sharing water and both the crops performed well and enhanced water use efficiency. Bio-irrigation is another beneficial phenomenon, which takes place in the intercrop combination of deep and shallow-rooted crops. Bio-irrigation is a hydraulic lift of the redistribution process of moisture. The deep-rooted crops penetrate taproot into the deeper layer of the soil to such water under moisture deficit conditions to cater for their physiological and metabolic activities and deposit a small quantity of water in the comparatively dry upper layer of the soil at night time when photosynthate production stops. The transported water to the upper layer of the soil is utilized by lateral secondary branches of the roots of the same crop and neighbouring crops also share it. The experimental results clearly indicated that deep-rooted pigeon pea shared played the role of bio-irrigators and shared moisture for shallow-rooted finger millet [67,68]. Moreover, Chai et al. [142] mentioned that WUE was increased by 95% in a maize + pea intercropping system over the pure stand of pea.

### 9.2.3. Atmospheric Carbon Dioxide

Plants use atmospheric carbon dioxide for production of assimilate or photosynthate by a biochemical process known as photosynthesis and release oxygen to the atmosphere. In an intercropping system sometimes more plant population is arranged and allowed for better utilization in space. In particular, in the additive series of intercropping, 100% of the population of the base crop is maintained as it remains in a pure stand and additional plant species are included as an intercrop with less population than the pure stand of the intercrop. In this way, the total population exceeds the population of the base crop in the pure stand and more plants exploit the available resources. If the crops and varieties are chosen properly, the complementarity effect is observed among the intercropped species

and more biomass is produced in this way as reported by earlier workers [2]. More biomass production is caused due to more assimilated production by the crops together and in this way, more quantity of greenhouse gas is used in the process [143–145].

### 9.2.4. Sunlight

Crops with a preferably dissimilar type of morphological characters are chosen in intercropping to assure complementarity among the species. Generally, for crops with shorter canopy structure, if selected with tall crops, dwarf species will certainly be affected by shade, but overall more sunlight will be used by the crops together in association. The combination of maize + cowpea was reported to enhance more light interception than sole cropping of maize [146]. Mahallati et al. [147,148] suggested that maize–bean strip cropping showed higher radiation absorption and system output than pure stands of either maize or bean. Sole cowpea and soybean used more photosynthetically active radiation (PAR) than intercropping when these legumes were intercropped with maize at different proportions, but maize + legume intercropping combination intercepted higher PAR over a pure stand of maize [101]. Experimental results indicated that defoliation of the top two leaves of maize at silking stage enhanced the productivity of maize + soybean intercropping system probably due to better light interception and partitioning of dry matter to reproductive parts [104].

### 9.2.5. Land Area

Generally, in the unit area, more plants of two or more crop species are sown in intercropping. The component crops may compete among themselves for land area. The component crops if differing in duration or canopy structure or morphological characters may show less competition [2] and preferably dissimilar crops are chosen in intercropping. To measure the efficiency of land area by intercrops, Willey and Osiru [86] proposed the concept of the LER. But interestingly if the component crops with similar durations may show their maximum need for growth resources almost at the same period and compete for the same. Therefore, crops with dissimilar growth habit are selected in intercropping to achieve less competition among species. In LER, the time factor or duration of the crops is not considered and that may be considered as a limitation of the expression to evaluate the efficiency of the intercropping system. Hiebsch [108] suggested another competition function named the area time equivalent ratio (ATER) where land area and time both are considered. For both the competition functions if the expression value exceeds unity (more than 1), the intercropping system is considered efficient.

### 9.3. Resource Conservation and Soil Health Management

Conservation of resources is also an important advantage of intercropping. Soil and water are efficiently managed and more coverage of ground area enhances the possibility of effective soil and water conservation. Furthermore, topsoil erosion is checked by more coverage of the ground area and thus loss of nutrients is also restricted.

### 9.3.1. Reduced Run-Off of Water, Soil Erosion and Nutrient Loss

In intercropping, the maximum ground area is covered; hence there will be a minimum chance of run-off, soil erosion and nutrient loss [123,148]. In an agroforestry system, *Gliricidia* alley cropping can reduce run-off by 28.2% and soil loss by 49.3–51.1% over no alley cropping system. Furthermore, *Gliricidia* alley can conserve soil organic carbon, N, P and K by 63.4, 5.0, 0.3 and 2.4 kg ha$^{-1}$. Similarly, the *Leucaena*-based alley cropping system is also effective in terms of checking run-off of water, conserving soil and preventing nutrient loss. *Leucaena* alley with a miniature trench can reduce run-off by 18.3–18.7% and soil loss by 37.2–43.0%. The alley cropping of *Leucaena* can conserve organic C, N, P and K by 57.7, 4.6, 0.3 and 2.2 kg ha$^{-1}$ [149]. The intercropping combination of finger millet + black gram recorded the lowest runoff (10.2%) and losses of soil and nitrogen,

phosphorus and potassium through erosion over sole when sown in contour because black gram covered enough ground area in intercropping with finger millet [150].

### 9.3.2. Soil Fertility Enhancement

The soil fertility status was also improved in finger millet + pulses intercropping which was due to contribution of leaf fall and biological nitrogen fixation by legumes [150]. The cereal–legume combination of intercropping is known to enable long term immobilization of N [151]. Nutrient balance studies indicated that cereal–legume intercropping enhanced N fertility of the soil. Among legumes, groundnut in combination with maize added N to the soil because of above- and below-ground architecture of groundnut and more soil coverage [152,153]. However, a study conducted at Basar, Arunachal Pradesh showed that a greater proportion of legumes in maize, either soybean or groundnut (1-row maize: 4 rows legume) enhanced P balance of the soil. However, irrespective of row proportion, legumes in intercropping with maize added K in the soil in the two years' trial [153]. The legume factor is responsible for such enhancement of soil fertility in a cereal legume intercropping system.

### *9.4. Sustainability*

The introduction of legumes as a component crop in the intercropping system can reduce the use of chemical inputs and thus minimize the emissions of greenhouse gases (GHGs). In the process of chemical N fertilizer production, $CO_2$ is generated. To meet the present need of chemical N fertilizers, annually 300 Tg of $CO_2$ is released to the atmosphere [15]. On the other hand, legumes fix N biologically and share the fixed N to non-legume crops in the mixture [48,154,155] and thus benefit not only the N economy of the crop production but also checking atmospheric pollution. Moreover, enhancement of above and below-ground diversity and change in pest population dynamics created a favourable environment for crop growth. Intercropping gives food and nutritional security to smallholders in drylands and natural insurance against crop failure. A lesser quantity of fertilizer is required in the legume-based intercropping system and thus use of outsourced chemicals in agriculture is reduced. Soil organic carbon (SOC) is a key factor for the enhancement of soil fertility and agricultural sustainability. In an intercropping system, total biomass production is obtained more than in sole cropping if the component crops are chosen wisely. More biomass production and biomass return in the form of leaf fall during the cropping period and stubble incorporation after harvest increase the organic content of the soil which is synonymous with SOC sequestration. All the benefits lead farming towards agricultural and environmental sustainability. Furthermore, the involvement of more labour inputs in intercropping assure engagement of family labourers including women in active participation in farming integrating gender equity and feminization [156,157] and an unemployed workforce in agriculture for stallholders [10] in the developing countries; thus, it ensures social sustainability.

### 9.4.1. Biotic Diversity

In the intercropping system, above-ground diversity is not only pronounced as the growing of more crops than one but also the crop mixtures increase the population of different arthropods, insects and birds. Omaliko [158] found a greater diversity of pollinators when cowpea was considered as a legume crop in intercropping. Furthermore, below-ground diversity is increased in the form of diversity in micro-organisms [22]. In legumes, when considered as a component crop in intercropping, the Rhizobium population is increased along with other beneficial micro-organisms like *Pseudomonas* sp., Alphaproteobacteria, Betaproteo bacteria and Cyanobacteria [159–161]. All these ultimately create a healthy ecosystem. Moreover, intercropping limits the population of harmful soil micro-organisms [84,162].

### 9.4.2. Food and Nutritional Security

For smallholders in developing countries, food and nutritional security is a huge task under subsistence farming. Mixture crops, especially, cereal and legume/oilseed combinations provide a large portion of the family requirement (calorie intake) and thus intercropping systems play a vital role in the alleviation of hunger.

### 9.4.3. Pest Population Dynamics

Intercropping systems can regulate the insects, diseases and weed population dynamics. Above ground, diversity is caused by the inclusion of two or more crops in intercropping. These crop mixtures attract pollinating bees and other predators which has a significant impact on production enhancement and insect population dynamics [2,22,163]. Intercropping of brassicas with various taxonomically unrelated crops increased the number of predators [164] and correspondingly reduced infestation of cabbage root fly and other Lepidopteran pests in comparison to the pure stand of brassicas [165]. When cowpea was grown in an intercropping system, the cotton sucking pest population was reduced [82] and green stink bug (*Nezara viridula*) and stem borer (*Chilo zacconius*) were checked when upland rice + groundnut was intercropped [83]. Intercropping also checks plant diseases. Actually, in intercropping a functional diversity is created which limits the population of harmful micro-organisms [84]. Growing of sorghum as an intercrop significantly reduced the incidence of bud necrosis disease of groundnut [166]. The following table (Table 4) indicates the restricted disease incidence in intercropping as evidenced by researchers. In intercropping, chemical herbicide application is difficult once crops emerge particularly when a combination of dicotyledonous and monocotyledonous plants is chosen in combination [76,167]. There is no reference available for applying certain types of pesticide to crop mixtures in the intercropping system. However, as the greater portion of the land area is covered by the crops in intercropping, there will be fewer weeds. Furthermore, allelopathy may reduce the weed population. Researchers noted less weed growth in different intercropping combinations like wheat and chickpea [71], maize + legume [72], and so on.

**Table 4.** Reduction of disease by the adoption of the intercropping system.

| Crop | Name of the Restricted Disease | Intercropping Combination | References |
|---|---|---|---|
| Potato | Bbacterial wilt (*Pseudomonas solanacearum*) | Maize + potato | [168] |
| Faba bean | Chocolate spot (*Botrytis fabae*) | Maize + faba bean and barley + faba bean | [169] |
| Beans | Angular leaf spot (*Phaeoisariopsis griseola*) | Maize + bean | [170] |
| Pea | Ascochyta blight (*Mycosphaerella pinodes*) | Cereal + pea | [171] |

### 9.4.4. Legume Effect and Less Chemical Fertilizers

The legume effect is pronounced as N benefits when these crops are considered in the intercropping system. Legumes are less N-demanding crops and these can fix atmospheric N biologically. The fixed N by legumes is used for their own nutrition and a portion is transferred to the associated non-legume in intercropping. A study on $^{15}$N labelling clearly indicated that N was transferred from soybean to corn when these crops were intercropped and seed inoculation of *Glomus mossacae* and *Rhizobium* facilitated the process [172]. Furthermore, P and K balance of the soil also increased due to the legume factor in cereal–legume association [153]. Such enhancement of soil fertility leads to less use of chemical fertilizers.

### 9.4.5. Crop Diversity and Natural Insurance

Biodiversity is enriched in intercropping and diversity in crop ecosystem assures sustainability. Under fragile ecological conditions in drylands, crop failure due to biotic and abiotic factors is a very common phenomenon and monoculture may be severely affected. But intercropping or polyculture is by nature diversified. As a result, total crop failure is much less likely and, thus, intercropping provides security and natural insurance to the farmers [2].

### 9.4.6. Ecosystem Services

The ecosystem is comprised of the biological community in the physical environment (all flora and fauna in the ecology) and their healthy interaction. When the ecology is congenial for proper nourishment of the biological community, healthy interaction will be observed. There is no doubt intercropping creates favourable ecological conditions in many ways to nurture the biological community as discussed earlier. Intercropping is beneficial for low C emission from the field and in a study it was noted that maize + pea and soybean + wheat emitted less C from the crop field [142]. Soybean minimizes $NO_3$ in the soil compared to other crops [173]. The presence of beneficial insects and micro-organisms and a lower population of weeds assures a healthy ecosystem in intercropping. Ecosystem services include provisioning good and services (food, forage and feed, biofuel and fuel), supporting services (pollination, biocontrol, C-sequestration, nutrient cycling, soil improvement), regulating and cultural services [174]. Among these, the first two ecosystem services are prominently observed in different types of the intercropping system adopted around the world.

### 10. Limitations of the Intercropping System

There are benefits of intercropping as stated earlier, but some limitations are also observed over mono-cropping (Table 5. These are mainly due to inter-species competition for limited resources, expression of allelopathic effects and difficulty in agronomic management of crops in the mixture. Generally, crops are chosen in such a way that they show complementarity or mutual sharing of nutrients, light and water and thus, advantages of intercropping are recorded [54,175]. However, if the choice of crop species is not appropriate, due to competition only the adverse effect may be noted in the productivity of crop mixture [176,177]. The selection of appropriate crops and suitable varieties, seeding rates and plant population, and manipulation in planting geometry of the crops can minimize inter-species competition among crops.

**Table 5.** Limitation of different intercropping system.

| Intercropping System | Limitation and Comments | References |
|---|---|---|
| Row intercropping | Preferably crops of dissimilar growth habits are grown to obtain higher level of complementarity and crops attain maturity at different times that make harvesting laborious. If crops are not chosen properly, inter-species competition may limit yields. | [48,54] |
| Mixed intercropping | Grass-legume is most common and harvested mainly as forage that creates no complexity and any limitation. But if crops are harvested separately, it will be labour intensive. | [54] |
| Relay intercropping | Succeeding crops may yield less compared to normal crops grown in sequential cropping and more seed rate of relay crop is required. | [178] |
| Strip intercropping | A combination of soil conserving and depleting crops are grown simultaneously in alternate strips. If perennial crops are grown in combination, may create shade problem to annuals. | [138] |

Plants release special bioactive chemicals (allelochemicals) which interrelate with the environment and both positive and negative impacts are observed. Allelopathy may be a negative issue in intercropping as allelochemicals produced by one species may hamper growth and productivity of another crop. For example, black walnut (*Juglans nigra* L.), a popular planted species in alley cropping, silvopastoral, and mixed-species systems, produces the chemical *juglone* that has an allelopathic effect on different crop species [179]. The aqueous leaf extracts of *Jatropha curcas* inhibits germination and retards shoot and root length in *Capsicum annum* L. [180].

A major disadvantage in intercropping is the difficulty in practical management of essential agronomic operations, particularly where farm mechanization is adopted or when the component crops grown in intercropping have dissimilar requirements for fertilizers, water and plant protection requirements. During or after harvest, mixed grains are separated which incur additional cost and, at harvest of the early maturing crop, there may be some mechanical disturbance to the long duration crop. Farm mechanization is really difficult in intercropping because machinery used for different faring operations like seeding, weed management, harvesting and threshing are made for big uniform fields. Further, during the harvest of one crop, there may be some kind of damage to other crops in combination. However, in intercropping of cereal + legume forage crops, there is no problem because both can be harvested or grazed at the same time [2]. In developing countries, the human workforce is available and farm mechanization is not fully adopted and under these conditions intercropping will not show any limitations. In many cases, it was further noted that intercropping caused yield reduction of the main/base crop than its pure stand because of competition among intercropped plants for light, soil nutrients and water [85]. This yield reduction may be economically meaningful if that particular main crop has a more attractive market price than the other intercropped plants. Furthermore, the intercropped canopy cover may cause a microclimate with a higher relative humidity conducive to disease incidence, especially of fungal pathogens [181].

## 11. Conclusions

Food and environmental security as well as enhancement of input use efficiency are global concerns in agriculture. Both the developed and developing nations are in a quest for a low carbon footprint in agriculture and thus there is an urgent need for a reduction of high energy chemical fertilizers, plant protection chemicals and energy use in farm mechanization. Furthermore, intensive agriculture caused a gradual degradation of natural resources and the enhancement of farm productivity is a tough job for all targeting future demand. Intensification of crops can be undertaken spatially and temporally by the adoption of the intercropping system. Intercropping ensures multiple benefits like enhancement of yield, environmental security, income as well as production sustainability and some ecosystem services. Among different ecosystem services, provisioning goods and services and supporting services are prominently observed in different types of intercropping system. In intercropping, two or more crop species are grown concurrently as they coexist for a significant part of the crop cycle and interact among themselves and with their related agro-ecosystem. Legumes as component crops in the intercropping system play versatile roles like biological N fixation and soil quality improvement, enhancement of environmental quality by reducing the use of chemical N fertilizer, additional yield output including protein yield, and creation of functional diversity. But growing two or more crops together requires additional care and management for the creation of less competition among the crop species and efficient utilization of natural resources. The choice of a proper intercropping system and appropriate management practices like the choice of crops, planting geometry, intercultural operation and plant protection are major concerns to obtain advantages from the intercropping system. The review provided an overview of earlier evidence indicating beneficial impacts of the properly managed intercropping system in terms of resource utilization and higher combined yield of crops grown with low inputs. In developing countries, resource-poor smallholders prefer to adopt low input

agriculture with the employment of the family workforce. Under subsistence farming where the production of sufficient food grain is a great challenge, beneficial impacts of the intercropping system are very common and proper utilization of resources by the adoption of intercropping ensures higher productivity as well as food security for a large number of smallholders in the world. Thus, the advantages of intercropping clearly derive from its usefulness as a low-input agriculture for food and environmental security in the present context.

**Author Contributions:** Conceptualization, S.M., A.H., T.S., P.B., J.B.P., J.J., U.B., S.K.D., S.L., and M.S. (Masina Sairam); writing—original draft preparation, S.M., A.H., T.S., P.B., J.B.P., J.J., U.B., S.K.D., S.L., and M.S. (Masina Sairam); writing—review and editing, A.H., M.S. (Milan Skalicky), P.O.; H.G.; M.B. and K.B.; funding acquisition, A.H., P.O., M.S. (Milan Skalicky); and M.B. All authors have read and agreed to the published version of the manuscript.

**Funding:** This research was funded by the 'Slovak University of Agriculture', Nitra, Tr. A. Hlinku 2,949 01 Nitra, Slovak Republic under the project 'APVV-18-0465 and EPPN2020-OPVaI-VA-ITMS313011T813'.

**Conflicts of Interest:** The authors declare no conflict of interest.

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
