# Peer review of "Intercropping—A Low Input Agricultural Strategy for Food and Environmental Security"

_agronomy, doi:10.3390/agronomy11020343_

Round 1

Reviewer 1 Report

This literature review is interesting and provides an appropriate synthesis on the subject 'intercroppping systems'. However, the first part of the introduction needs to be reworked to better trace the historical evolution of this cultivation practice.

I invite the authors to develope the question of the limits of intercropping systems, particularly in relation to the typology proposed in section 4. For example, the authors can make a table listing the different limits according to the type in question.

The thesis that appeared in the title is not necessarily demonstrated in the text. The conclusion takes up some ideas in this sense but remains insufficient to make the link between the intercropping system and food and environmental security.

Author Response

Response to Reviewer_1 comments

Dear Sir/ Madam,

We must appreciate for your valuable comments and revision has been made accordingly.

  • Your comment: This literature review is interesting and provides an appropriate synthesis on the subject 'intercropping systems'. However, the first part of the introduction needs to be reworked to better trace the historical evolution of this cultivation practice.

Authors’ response: Done, historical evidences included

  • Your comment: I invite the authors to develop the question of the limits of intercropping systems, particularly in relation to the typology proposed in section 4. For example, the authors can make a table listing the different limits according to the type in question.

Authors’ response: Done as per your suggestions. A table (Table 6) has been made.

  • Your comment: The thesis that appeared in the title is not necessarily demonstrated in the text. The conclusion takes up some ideas in this sense but remains insufficient to make the link between the intercropping system and food and environmental security.

Authors’ response: Addressed and linked as per your comments

Reviewer 2 Report

The topic is relevant but the text needs an extensive revision of English
as it is difficult to understand, the text is very long and there is a lack
of precision in the definition of some terms, such as "sustainability".
What sustainability? environmental, economic, social?
References are missing at points where they would be very important,
especially if this is a review article.
The conclusion is poor for such a large article. I suggest the extensive review of the text focusing on these points.

Author Response

Response to Reviewer_2 comments

Dear Sir/ Madam,

We are really grateful to you for your kind comments for upgradation of the review article and revision has been made accordingly.

  • Your comment: The topic is relevant but the text needs an extensive revision of English as it is difficult to understand, the text is very long and there is a lack of precision in the definition of some terms, such as "sustainability". What sustainability? Environmental, economic, social?

Authors’ response:

  1. English language has been checked and modified accordingly.
  2. Mainly agricultural sustainability is addressed. But there are direct and indirect impacts of intercropping on environmental (improvement of soil health, reduction of chemical inputs, better utilization of available natural resources), economic (yield enhancement from unit area, natural insurance against crop failure under harsh climate and more income) and social (engagement of under-employed family labourers under smallholders’ conditions and greater yield and income) sustainability and these have been mentioned.
  • Your comment: References are missing at points where they would be very important, especially if this is a review article.

Authors’ response:  You are absolutely right and references Included.

  • Your comment: The conclusion is poor for such a large article. I suggest the extensive review of the text focusing on these points.

Authors’ response: Revised and conclusion part elaborated.

Reviewer 3 Report

Dear authors, please find my 21 comments within the attached annotated PDF version of the manuscript. Kind regards.

Author Response

Response to Reviewer_3 comments

Dear Sir/ Madam,

We are really happy to say that you have reviewed the article thoroughly and made your valuable comments for qualitative improvement. We are really grateful to you.

As per your comments (in pdf version with sticky note), we have made the revision of all points. The revisions made have been mentioned as per the line number (of your comments; pdf version) and these are as follows.

Your comments and authors’ response

Line 2: Title has been changed and the word ‘system’ has been removed.

Line 43: As per the suggestion, the words have been changed.

Line 60: The sentence has been deleted.

Line 88: Defined: “Modern agriculture based on supply with high energy and fossil-fuel based inputs those commonly known as Green Revolution Technologies”.

Line 178-179: Reference of von Cossel et al. (2020) and Weißhuhn et al. (2017) included. One more reference cited.

Line 221-222: Von Cossel et al. 2019 reference cited.

Line 284-293: References cited.

Line 441: Thanks a lot sir for your valuable remarks.

Line 553-554: As per your advice, modified and references included.

Line 575: Changed. It’s “agriculture based on intercropping”. Thanks a lot.

Line 602: Reference included. Thanks.

Line 717-731: References added.

Line 760: Addressed as per the comments. Reference added.

Line 777: Changed. Thanks a lot.

Line 793: References added.

Line 828: Income sustainability added. Ecosystem services explained in next sentence.

Line 830: Addressed as per your valuable comment.

All corrections and revisions have been made and now we are looking forward for your necessary steps.
